# Ameliorative Effects of *Bifidobacterium animalis* subsp. *lactis* J-12 on Hyperglycemia in Pregnancy and Pregnancy Outcomes in a High-Fat-Diet/Streptozotocin-Induced Rat Model

**DOI:** 10.3390/nu15010170

**Published:** 2022-12-29

**Authors:** Jianjun Yang, Yumeng Ma, Tong Li, Yuanxiang Pang, Hongxing Zhang, Yuanhong Xie, Hui Liu, Yanfang Sun, Jianhua Ren, Junhua Jin

**Affiliations:** 1Beijing Laboratory of Food Quality and Safety, Beijing Key Laboratory of Detection and Control of Spoilage Organisms and Pesticide Residues in Agricultural Products, Food Science and Engineering College, Beijing University of Agriculture, Beijing 102206, China; 2Beijing Advanced Innovation Center for Food Nutrition and Human Health, College of Food Science & Nutritional Engineering, China Agricultural University, Beijing 100083, China; 3Laboratory of Precision Nutrition and Food Quality, Ministry of Education, Department of Nutrition and Health, China Agricultural University, Beijing 100083, China; 4State Key Laboratory of Feed Microbial Engineering of Dabei Agriculture, Beijing 100190, China; 5Ningxia Saishang Dairy Co., Ltd., Yinchuan 750299, China; 6College of Bioengineering, Beijing Polytechnic, Beijing 100176, China

**Keywords:** *Bifidobacterium*, hyperglycemia, pregnancy, physiology, histopathology, fetuses, placental microbiota

## Abstract

Bifidobacterium, a common probiotic, is widely used in the food industry. Hyperglycemia in pregnancy has become a common disease that impairs the health of the mother and can lead to adverse pregnancy outcomes, such as preeclampsia, macrosomia, fetal hyperinsulinemia, and perinatal death. Currently, Bifidobacterium has been shown to have the potential to mitigate glycolipid derangements. Therefore, the use of Bifidobacterium-based probiotics to interfere with hyperglycemia in pregnancy may be a promising therapeutic option. We aimed to determine the potential effects of *Bifidobacterium animalis* subsp. *lactis* J-12 (J-12) in high-fat diet (HFD)/streptozotocin (STZ)-induced rats with hyperglycemia in pregnancy (HIP) and respective fetuses. We observed that J-12 or insulin alone failed to significantly improve the fasting blood glucose (FBG) level and oral glucose tolerance; however, combining J-12 and insulin significantly reduced the FBG level during late pregnancy. Moreover, J-12 significantly decreased triglycerides and total cholesterol, relieved insulin and leptin resistance, activated adiponectin, and restored the morphology of the maternal pancreas and hepatic tissue of HIP-induced rats. Notably, J-12 ingestion ameliorated fetal physiological parameters and skeletal abnormalities. HIP-induced cardiac, renal, and hepatic damage in fetuses was significantly alleviated in the J-12-alone intake group, and it downregulated hippocampal mRNA expression of insulin receptor (*InsR*) and insulin-like growth factor-1 receptor (*IGF-1R*) and upregulated *AKT* mRNA on postnatal day 0, indicating that J-12 improved fetal neurological health. Furthermore, placental tissue damage in rats with HIP appeared to be in remission in the J-12 group. Upon exploring specific placental microbiota, we observed that J-12 affected the abundance of nine genera, positively correlating with FBG and leptin in rats and hippocampal mRNA levels of *InsR* and *IGF-1R* mRNA in the fetus, while negatively correlating with adiponectin in rats and hippocampal levels of *AKT* in the fetus. These results suggest that J-12 may affect the development of the fetal central nervous system by mediating placental microbiota via the regulation of maternal-related indicators. J-12 is a promising strategy for improving HIP and pregnancy outcomes.

## 1. Introduction

Hyperglycemia in pregnancy (HIP) is characterized by an increase in blood sugar caused by impaired blood glucose metabolism due to factors such as improper diet, obesity, and elevated hormone levels in pregnant women, mainly pregestational and gestational diabetes mellitus (PGM and GDM, respectively) [1]. HIP is considered a major underlying cause of pregnancy-related complications and a contributing cause to health risks throughout the subsequent life of both mothers and fetuses, with amplification of the global epidemic of non-communicable diseases [2]. Although the pathogenesis of HIP has not been comprehensively elucidated, previous studies have shown that HIP might be related to maternal hormone imbalances and insulin resistance [3,4]. In addition, growing evidence suggests that insulin or drug therapy may contribute to the risk of adverse pregnancy outcomes, such as premature birth, hypoglycemia, and macrosomia in newborns [5,6,7]. Thus, efforts have been made to explore more effective strategies to prevent or treat HIP.

Probiotics are living microorganisms that confer health benefits to the host [8,9], affording possible improvements in metabolic disorders during pregnancy [10]. Several studies have indicated that probiotic-mediated modulation of the gut microbiota might alleviate increased blood glucose levels, insulin resistance, inflammatory markers, and oxidative stress caused by HIP [11,12,13]. In a GDM animal model, probiotics could significantly improve fasting blood glucose (FBG) in GDM and alter the gut microbiota after probiotics, as determined by 16S rRNA sequencing [14]. However, few reports have examined the effects of probiotics on the fetuses of females with HIP.

The placenta, as a major link between the mother and fetus, plays a vital role in fetal growth and development during pregnancy and utilizes its own complex vascular system to exchange different substances between the mother and the fetus via circulating blood [15,16]. Recent studies have shown that the placenta is not a “sterile” organ but possesses its own endogenous microbiota [17]. Reportedly, the placental microbiota is mainly composed of non-pathogenic symbiotic microorganisms, most similar to the human oral microbiota, closely related to human health and pregnancy outcomes [18,19]. However, the effects of bioactive substances on placental microbiota remain unknown.

*Bifidobacterium* is a common probiotic that is widely used in the food industry [20,21]. *Bifidobacterium animalis* subsp. *lactis* J-12 (J-12), isolated from the intestinal tract of breastfed 3-month-old infants, was found to survive in the intestinal environment for 36 h and remain in the intestinal tract for 7 days after one-time ingestion for 2 weeks and relieve constipation in humans [22,23]. Moreover, we previously reported that J-12 can inhibit glucose production and hinder glucose transport, thereby reducing blood glucose levels and alleviating insulin resistance [22]. However, the impact of this strain on HIP remains unclear.

In the present study, HIP was induced by using a high-fat diet (HFD)/streptozotocin (STZ) in rats. We then explored the effects of J-12 on the fetal development of the physiological, histopathological, and central nervous systems. Furthermore, we analyzed the correlation between maternal and fetal alterations by exploring placental histopathology and microbiota.

## 2. Materials and Methods

### 2.1. Strain and Preparation of Bacterial Suspension

*B. animalis* subsp. *lactis* J-12 (CGMCC NO.25005), which was stored in the Beijing Laboratory of Food Quality and Safety, Beijing University of Agriculture (Beijing, China), was inoculated into MRS liquid culture medium. The third-generation culture of J-12 was centrifuged (8000× *g*, 10 min, 4 °C) and washed with phosphate-buffered saline (PBS; pH 7.4), and the concentration was adjusted to 10^9^ CFU/mL for further experiments.

### 2.2. Chemicals and Drugs

STZ was obtained from Sigma-Aldrich (St. Louis, MO, USA), and insulin (Lantus 100 U/mL) was obtained from Sanofi Pharmaceutical Co., Ltd. (Beijing, China). 

### 2.3. Animal Experimental Design

All experimental animal procedures were approved by the Ethical Committee of the Experimental Animal Care of Beijing University of Agriculture (Beijing, China). A total of 100 (4-week-old) virgin female Sprague Dawley rats, acquired from Beijing Weitong Lihua Laboratory Animal Technology Co., Ltd. (Beijing, China), were housed in standard cages (4 rats per cage) and maintained under standard laboratory conditions: 22 ± 2 °C, 50 ± 10% relative humidity and a 12 h/12 h light/dark cycle. Rats were provided food and water ad libitum for seven days while adapting to the new environment.

Then the rats were randomly divided into five groups: control group (CK), model group (M), insulin group (I), J-12 group (J-12), and J-12 plus insulin group (J + I). The animals assigned to the control group were fed a standard diet (D12450B, containing 10% kcal from fat, Beijing Keao Xieli Feed Co., Ltd., Beijing, China), while those in the other groups were fed an HFD (D12492, containing 60% kcal from fat, Beijing Keao Xieli Feed Co., Ltd., Beijing, China) for 1 to 14 weeks. After feeding the HFD for 14 weeks, the rats were fasted for 12 h overnight and subsequently administered a single STZ injection (40 mg/kg, dissolved in 0.1 mM cold citrate buffer; pH 4.4) intraperitoneally to induce hyperglycemia (Nath, Ghosh, and Choudhury 2017). The CK group was injected with citrate buffer. Therefore, all groups were fed a standard diet, and blood glucose was monitored in real time for 15 to 17 weeks for stabilization. 

Female rats with normal FBG levels (<6.1 mmol/L) in the CK group were selected for mating with male rats (female:male = 1:1). Meanwhile, female rats with FBG levels > 6.1 mmol/L in the other groups were selected for mating with male rats (female:male = 2:1) to establish the HIP model. Female and male rats were mated overnight, vaginal plugs were observed, vaginal smears were collected the following morning, and pregnancy was verified by the presence of plugs and spermatozoa, indicating the first day of gestation (GD0). All pregnant rat groups were fed a standard diet, and the I and J + I groups were subcutaneously administered insulin at 14 IU/kg daily. 

It must be stated that the rats in the J-12 and J + I groups were administered the J-12 suspension (1 × 10^9^ CFU daily) by gavage during the entire experimental period, and the other three groups were administered the same amount of PBS. FBG and glycosylated hemoglobin (HbA1c) levels were monitored throughout the modeling process.

### 2.4. Analysis of Relevant Parameters in Rats with HIP

#### 2.4.1. Determination of FBG and HbA1c Levels

Rats with HIP were subjected to a 12 h fast. Blood was collected from the tail vein, and FBG levels were measured by using a blood glucose meter (Roche Diagnostics, Penzberg, Germany) at GD 1, 5, 10, 14, and 19. 

HbA1c levels were measured by using a glycosylated hemoglobin detector (Bohuisi Biomedical Technology Co., Ltd., Jiangsu, China) at GD 1, 10, and 19.

#### 2.4.2. Assessment of Serum Biochemistry

At the end of the GD 20 experiment, 10% chloral hydrate was injected intraperitoneally into rats with HIP. After anesthesia induction, blood samples were collected from the abdominal aorta and centrifuged at 4000× *g* and 4 °C for 10 min. The serum was collected and frozen at −80 °C for subsequent assays. Enzyme-linked immunosorbent assay kits were used to measure triglyceride (TG; Code No. A110-2-1), total cholesterol (TCHO; Code No. A111-2-1), leptin (Lep; Code No. H174-1-1), adiponectin (ADPN; Code No. H179-1-1), and insulin (Ins; Code No. H203-1-1), according to the manufacturer’s protocol, at the Nanjing Jiancheng Bioengineering Institute.

#### 2.4.3. Histological Analysis

The pancreatic and hepatic tissues of rats with HIP were fixed in 4% formaldehyde (Beijing Changhua Zhicheng Technology Co., Ltd., Beijing, China) and embedded in paraffin for hematoxylin–eosin (H&E) staining, using standard protocols and histological scoring criteria [24,25].

### 2.5. Analysis of Relevant Parameters in Fetuses 

#### 2.5.1. Experimental Evaluation of Fetuses

The rats with HIP were euthanized at GD20, and fetuses were obtained by Cesarean section. The numbers of live and dead fetuses and resorptions were recorded and weighed.

#### 2.5.2. Skeletal Assessment of Fetuses

To assess the fetal skeletal development, half of the obtained fetuses were fixed in 95% (*v*/*v*) ethanol for at least 15 days, as described by Staples [26].

#### 2.5.3. Histological Analysis 

Fetal cardiac, renal, and hepatic tissues were fixed with 4% formaldehyde and embedded in paraffin for H&E staining, using standard protocols and histological scoring criteria [23,24].

#### 2.5.4. Transmission Electron Microscopy (TEM) Analysis

Fetal cardiac and renal tissues were treated as previously described [27]. In short, tissues were fixed in 2.5% mL Eppendorf (EP) tubes^®^ containing precooled 1.5 glutaraldehyde for ≥2 h, rinsed, dehydrated, embedded, solidified, and sectioned (70 nm) for TEM analysis.

#### 2.5.5. RNA Extraction and Quantitative Reverse Transcription-Polymerase Chain Reaction (qRT-PCR)

Fetal hippocampi were collected and stored at −80 °C until further analysis. Briefly, frozen hippocampi were extracted in an enzyme-free EP tube (1.5 mL) with precooled Trizol lysate (Ambion, Austin, TX, USA), according to the manufacturer’s instructions. Complementary DNA (cDNA) was obtained by reverse transcription, using the PrimeScript™ RT Reagent Kit with gDNA Eraser (Perfect Real Time) (TaKaRa, Kyoto, Japan, Code. No. RR047A). The oligonucleotides used are listed in Table 1. The qRT-PCR conditions were as follows: 30 s at 95 °C; 40 cycles of 5 s at 95 °C, 34 s at 56 °C or 60 °C, and 30 s at 72 °C, followed by a melting curve step. The mRNA abundance was calculated according to the 2^−ΔΔCt^ method, using glyceraldehyde-3-phosphate dehydrogenase (GAPDH) or β-actin as an internal control (reference gene), and expressed as a percentage of the control group induction (calibrator) [28].

### 2.6. Analysis of Relevant Placental Parameters 

#### 2.6.1. Placental Sample Collection 

Placental samples were collected as previously described [18]. The personnel wore face masks and gloves to ensure sterility. Then placentas were fixed by using 4% formaldehyde or frozen at −80 °C for further analysis.

#### 2.6.2. Placental Histological 

Placental samples were fixed by using 4% formaldehyde and embedded in paraffin for H&E staining, using the experimental method and histological scoring criteria described by Hosni et al. [25].

#### 2.6.3. Placental Microbial Diversity

Placental samples were frozen in liquid nitrogen and stored at −80 °C until use. DNA was extracted by using a FastDNA^®^ Spin Kit for Soil (MP Biomedicals, Solon, OH, USA, No. 6560200). DNA from each sample was used to amplify the V3 and V4 regions of the 16S rRNA gene. The PCR conditions were as follows: 95 °C for 3 min; 29 cycles of 95 °C for 30 s, 53 °C for 30 s, 72 °C for 45 s, followed by 72 °C for 10 min. The PCR mixtures contained 4 μL of 5 × TransStart FastPfu buffer, 2 μL of 2.5 mM dNTPs, 0.8 μL forward primer (5 μM), 0.8 μL reverse primer (5 μM), 0.4 μL TransStart FastPfu DNA Polymerase, 10 ng template DNA, and ddH_2_O up to 20 μL. PCRs were performed in triplicate. The PCR products were extracted in a 2% agarose gel and purified by using an AxyPrep DNA Gel Extraction Kit (Axygen Biosciences, Union City, CA, USA), according to the manufacturer’s instructions, followed by quantification using a Quantus™ Fluorometer (Promega, Madison, WI, USA).

Purified amplicons were pooled in equimolar amounts and paired-end sequenced on an Illumina MiSeq PE300 platform/NovaSeq PE250 platform (Illumina, San Diego, CA, USA), according to the standard protocols of Majorbio Bio-Pharm Technology Co., Ltd. (Shanghai, China). Raw reads were deposited in the NCBI Sequence Read Archive database (Accession Number: SRP374016). Operational taxonomic units (OTUs) with a 97% similarity cutoff were clustered by using UPARSE version 7.1, and chimeric sequences were identified and removed [29,30]. The taxonomy of each OTU representative sequence was analyzed by RDP Classifier version 2.2 against the 16S rRNA database, using a confidence threshold of 0.7 [31]. Finally, we used linear discriminant analysis effect size (LEfSe) to determine taxa that most likely explain the differences between experimental and control samples [32]. Simultaneously, we performed a correlation analysis between environmental factors and placental microbiota.

### 2.7. Statistical Analysis

Data are shown as mean ± standard deviation (SD). GraphPad prism (version 9.0) was employed to analyze most data, using one-way analysis of variance (ANOVA) with Tukey’s multiple comparison test. A significance analysis of the percentages was performed by using the chi-square test. A *p*-value of < 0.05 was deemed a significant difference between groups. Histological H&E-stained sections were analyzed by using the 2.4 version of CaseViewer scan browsing software. The placental microbial diversity was analyzed by using the I-Sanger cloud platform. For biochemical parameters, *n* = 6 per group was employed, whereas *n* = 3 per group was used for histopathological analysis.

## 3. Results

### 3.1. FBG and HbA1c Levels during Pregnancy in Rats

FBG levels on GD 1, 5, 10, 14, and 19 were monitored continuously. The FBG levels in the CK group were relatively stable (<6.1 mmol/L), while levels in other groups fluctuated between 20 and 30 mmol/L (Figure 1A). In late pregnancy (GD 19), FBG levels were significantly higher in the M group (29.07 mmol/L) than in the CK (4.48 mmol/L), I (24.67 mmol/L) and J-12 (26.00 mmol/L) groups (Figure 1B). In contrast, the FBG level was significantly lower in the J + I group (21.20 mmol/L) than in the M group (Figure 1B). Therefore, it can be speculated that J-12 cooperated more effectively with insulin to relieve HIP. Next, we assessed HbA1c levels on GD 1,10, and 19. Although HbA1c levels fluctuated in all groups, levels in the HFD/STZ-induced groups were persistently higher than 6.50%, which was significantly higher than that documented in the CK group (5.40%) (Figure 1C–E). Based on these findings, we established an animal model of HIP.

### 3.2. Effect of J-12 on Dyslipidemia in Rats with HIP

Intervention with J-12 significantly lowered serum triglycerides (TGs), and significant differences were observed in the M group (12.66 mmol/L), I group (13.7 mmol/L), and J + I group (12.92 mmol/L) (Figure 2A). Compared with the M group (7.99 mmol/L), the TCHO level in all other groups was significantly decreased, and the TCHO levels were 3.52 mmol/L and 4.00 mmol/L in the J-12 and J + I group, respectively.

### 3.3. Effect of J-12 on Insulin, Leptin, and Adiponectin Levels in Rats with HIP 

Notably, compared with the CK group (1.27 mIU/L insulin, 10.69 ng/mL leptin, and 14.57 mg/L adiponectin), serum levels of insulin and leptin were significantly elevated in the M group (16.36 mIU/L, 14.02 ng/mL), whereas the adiponectin level (9.96 mg/L) was significantly reduced, indicating HFD/STZ-induced insulin and leptin resistance in rats with HIP (Figure 3). Furthermore, J-12 improved insulin and leptin resistance (1.00 mIU/L, 9.16 ng/mL) and activated adiponectin (14.97 mg/L) in rats with HIP, with no significant difference detected compared with the CK group (Figure 3). In addition, insulin levels in the I (41.30 mIU/L) and J + I (24.42 mIU/L) groups were significantly increased; however, the insulin level was significantly lower in the J + I group than in the I group (Figure 3A). 

### 3.4. Effect of J-12 on Pancreatic and Hepatic Tissue Damage in Rats with HIP

In the CK group, H&E analysis of pancreatic tissues revealed that islets were normally distributed (double arrows), and the interlobular ducts were well developed (single arrows) and distributed between pancreatic lobules, without cell degeneration, granular degeneration, glycogen precipitation, or inflammatory cell infiltration (Figure 4A). In contrast, the M group exhibited irregularly shaped degenerated islets (double arrows), irregular lobular ducts (single arrows), and a small amount of inflammatory cell infiltration, whereas the I, J-12, and J + I groups partially recovered islet cells and showed alleviated inflammatory cell infiltration (Figure 4A).

Consistently, H&E-stained liver tissue sections displayed a clear hepatolobular structure and the neatly arranged hepatocyte cords in the CK group (Figure 4B). The hepatic sinusoids exhibited no obvious stenosis or expansion swelling, with no notable inflammatory cells (Figure 4B). Compared with the CK group, liver cells in the M group were swollen (black arrow), enlarged in volume, slightly loose in part of the cytoplasm, accompanied by scattered inflammatory cell infiltration (yellow arrow) in hepatic sinusoids and some portal areas, fibrous connective hyperplasia (blue arrow), and slight bile duct cell proliferation (green arrow) in some portal areas (Figure 4B). Intervention with J-12 significantly reduced hepatic tissue inflammation, exhibiting a morphology similar to that of the CK group (Figure 4B). Histopathological scores of pancreatic and hepatic tissues are shown in Appendix A.

### 3.5. Effect of J-12 on Physiologic Parameters of Fetuses

The physiological parameters of the fetuses are listed in Table 2. The rate of survival was significantly higher in the CK group than in the M group; however, fetal mortality was significantly higher in the M group than in other groups. No significant difference was noted between the J-12 and CK groups in the rate of survival, along with no dead or resorbed fetuses. In contrast, the resorption rates (2.6% and 4.8 %, respectively) were higher in the I and J + I groups than in the control group. No significant alterations in fetal body weight were observed.

### 3.6. Effect of J-12 on Fetal Skeletal Abnormalities

Statistically significant differences (*p* < 0.05) in the rate of fetuses with skeletal anomalies were observed among experimental groups. Specifically, the M, I, and J-12 groups demonstrated the absence of the xiphoid process and sternal fractures when compared with the CK group (*p* < 0.05). A lower number of absent xiphoid processes were noted in fetuses of the J-12 group than in the M, I, and J + I groups (*p* < 0.05). In the J + I group, fetuses showed an increased rate of unossified coccyges and incomplete ossification of the bottom vertebra (*p* < 0.05) (Table 3).

### 3.7. Effect of J-12 on Fetal Tissue Damage 

#### 3.7.1. Effect of J-12 on Fetal Cardiac Tissue Damage 

In addition, fetal cardiac tissues were histopathologically evaluated. Cardiac tissues in the CK group showed obvious mitosis with regular and transverse bands of cardiomyocytes, while those in the M group displayed nuclear consolidation, cytoplasmic vacuolization, hydrological changes, irregular boundaries, reduced cytoplasmic staining intensity, and no obvious mitosis (Figure 5A). Similarly, we observed that cardiac organelles in the CK group were structurally intact and well-defined, as determined by using TEM. In the M group, cell organelles were blurry, nuclei were shrunken and irregular in shape, and chromatin was clustered (Figure 5B). However, the related pathological morphology and TEM results of fetal cardiac tissues were alleviated in the I, J-12, and J + I groups when compared with the M group (Figure 5A,B). The histopathological scoring of cardiac tissue is shown in Appendix A.

#### 3.7.2. Effect of J-12 on Fetal Renal Tissue Damage 

In the CK group, H&E-stained sections of fetal renal tissues presented intact renal corpuscles, consisting of a renal capsule (dashed arrow) and a mature glomerulus erythrocyte space (double-headed arrow), as well as proximal and distal convoluted tubules. Glomerular shrinkage, red blood cell space (red double arrow), tubular vacuole formation, proximal tubule brush border detachment, and distal tubule epithelial degeneration were observed in the M group. Compared with the M group, a slight increase in the red blood cell space (red double arrow), a small amount of degeneration of the proximal and distal convoluted tubules, and a small vacuolar area were observed in the I and J-12 groups; in general, the tissue morphology was similar to that of the CK group. In addition, a small number of glomeruli and vacuolated areas were noted in renal tubules exhibiting intact glomeruli, renal capsule (dashed arrow), preserved erythrocyte space (red double arrow), intact proximal convoluted tubules, and distal convoluted tubules in the J + I group (Figure 5C). Moreover, further observation of glomerular changes, using TEM, revealed that the glomeruli in the CK group showed an intact glomerular capillary lumen and formed a thin fenestrated layer within the basement membrane, with intact podocytes and foot processes surrounding the capillaries. In contrast, the M group exhibited a thickened glomerular basement membrane, endothelial cells with no obvious fenestrated layer, shrunken nuclei, and widened foot processes with a diffuse fusion tendency. However, glomerular pathological morphology was relieved in the other three groups. Glomerular damage was alleviated in the I, J-12, and J + I groups (Figure 5D). The histopathological scoring of renal tissue is shown in Appendix A.

#### 3.7.3. Effect of J-12 on Fetal Hepatic Tissue Damage 

The arrangement of the hepatic cords was slightly disordered, and a small number of hematopoietic and red blood cells were noted in the hepatic sinusoids in the CK group. Compared with the CK group, the hepatic cord was broken (black arrow), the structure was disordered, and there were relatively more hematopoietic cells in the hepatic sinusoid (blue arrow) in the M group. Following intervention with insulin, J-12, and combination therapy with J-12 and insulin, the degree of hepatic tissue damage was reduced when compared with that in the M group, and the morphology was similar to that of the CK group (Figure 5E). The histopathological scoring of the hepatic tissue is shown in Appendix A. In summary, J-12 could prevent hepatic function damage in fetuses of rats with HIP. 

### 3.8. Effect of J-12 on Gene Expression in the Central Nervous System of Fetuses

We examined the effects of treatment interventions on the fetal expression of insulin receptor (InsR), insulin-like growth factor-1 receptor (IGF-1R), and protein kinase B (AKT) in the hippocampus on postnatal day 0 (P0). Compared with the other groups, hippocampal expression of *InsR* and *IGF-1R* mRNA was significantly upregulated in the M group. No significant difference was observed between the I and J-12 groups, while the J + I group showed significantly reduced hippocampal expression of *InsR* and *IGF-1R* mRNA (Figure 6A,B). In contrast, the hippocampal expression of *AKT* mRNA was significantly reduced in the M group when compared with that in other examined groups. We found that, although there was a significant difference between the CK and J-12 groups, the J-12 group was more similar to the CK group than other groups. Additionally, the hippocampal expression of *AKT* mRNA was significantly higher in the I and J + I groups than in the M group (Figure 6C).

### 3.9. Effect of J-12 on Placental Tissue Damage in Rats with HIP

Human and rodent placentas are villous placentas, mainly comprising syncytiotrophoblast and cytotrophoblast cells. These cell layers form the maternal–fetal barrier of the villous tissue, thus effectively regulating the transfer and circulation of nutrients from the mother to the fetus [33]. Understanding placental morphology is essential to clarify its pathophysiology. Accordingly, we examined the structural layer of the placenta after H&E staining (Figure 7 and Appendix A). In the CK group, the structure of each layer of placental tissue was clear and divided into the basal (including decidua and trophoblast layer) and labyrinth layers. Macrophage lysis and sponge trophoblast necrosis were not observed in trophoblast layers. There were abundant villi in the labyrinth layer, with no obvious damage to trophoblasts, as well as no obvious extrusion of fetal blood vessels. The decidual area of the placental tissue was thinner in the M group than in the CK group, while the trophoblast was significantly thickened, and the number of glycogen cells increased. In addition, we observed a large number of degenerated glycogen cells (shown with black arrows), a significantly reduced number of sponge trophoblasts, edema (green arrows), individual degeneration (red arrows), atrophic and fractured labyrinthine layer (blue arrows), loose cytoplasm, markedly reduced fetal blood vessels in villi (brown arrows), and blood sinusoids filled with red blood cells. Thus, after the intervention with insulin, J-12, and combination therapy with J-12 and insulin, the symptoms were relieved, and the histological morphology tended to be similar to that of the CK group, indicating that the treatment strategies alleviated the damaged placental structure.

### 3.10. Effect of J-12 on Placental Community Diversity in Rats with HIP

To explore whether J-12 impacted the placental microbiota, 1,004,324 high-quality sequences of the V3–V4 region of 16S rRNA were collected from 22 fecal samples (5 groups; CK group, *n* = 6; other groups, *n* = 4 per group), with an average sequence length of 427 bp. The alpha diversity (α-diversity) of placental microbiota in each group was analyzed at the genus level (Table 4). Compared with the M group, the Sobs index (*p* < 0.01), ace index (*p* < 0.01), and Chao index (*p* = 0.01) of the J-12 group were significantly reduced, indicating that J-12 affected the α-diversity of the placental microbiota. 

Proteobacteria, Firmicutes, Actinobacteria, and Bacteroidetes were the four major bacterial phyla detected in all groups (Figure 8A–E). The M group presented a decreased proportion of Proteobacteria and an increased proportion of Firmicutes (69.96%, 25.51%) when compared with the CK (87.04%, 3.96%) group (Figure 8B,C). Meanwhile, compared with the M group (69.96%, 25.51%), the proportion of Proteobacteria increased, and that of Firmicutes decreased in the J-12 group (75.64%, 17.61%) (Figure 8B,C). In addition, a significant increase in Actinobacteria was detected in the J + I group (21.20%) (Figure 8D). Simultaneously, we analyzed placental microbiota at the genus and species levels and found that *Delftia* in the phylum Proteobacteria, especially *Delftia_tsuruhatensis*, was the dominant species in the placental microbiota structure (Figure 8F,G). Compared with the other groups, the M group presented a relatively high proportion of *Weissella_cibaria*. The relative proportion of *Corynebacterium_glutamicum* was significantly higher in the J + I group than in the other groups (Figure 8G).

The LEfSe method was used to analyze the crucial species that were significantly altered following treatments. The threshold of the logarithmic LDA score of 3.5, which was used to distinguish features. According to the size of the LDA score (Figure 8H), the M group was characterized by higher amounts of Firmicutes and Bacilli, while Nakamurellaceae and Chishuiella were significantly enriched in the J-12 group, and the enrichment of Actinobacteria and Corynebacteriales was notably enhanced in the J + I group.

### 3.11. Predicted Metabolic Profile of the Placental Microbiome after J-12 Supplementation

In view of the above structural changes, the 16S rRNA data were further analyzed by using PICRUSt to predict the metabolism of fecal microbiota in all groups. The Kyoto Encyclopaedia of Genes and Genomes database (KEGG, http://www.genome.jp/kegg/ (accessed on 10 November 2022)) was used for determining whether J-12 caused obvious changes in the functional pathways of the placental microbiota in rats with HIP.

The J-12 could regulate metabolism and the genetic- and environmental-information-processing-related pathways (Figure 9). The metabolic pathways are involved in glucose- and lipid-metabolism-related pathways. Among these, glucose-metabolism-related pathways, carbon metabolism, biosynthesis of amino acids, glyoxylate and dicarboxylate metabolism, pyruvate metabolism, and oxidative phosphorylation were characterized. The lipid-metabolism-associated pathway includes the most critical fatty acid metabolism. The genetic information processing pathways involved in ribosome and amino acid changes were characterized. The environment-information-processing pathway involved ABC transporters (membrane transport). In summary, functional enrichment suggests different preferences in each group, but it is dominated by metabolic-pathway-related functions.

### 3.12. Correlation between Overall Placental Microbiota Structure and Maternal and Fetal Rat Parameters

Our next goal was to identify specific placental bacteria potentially associated with certain maternal and fetal rat parameters. At the genus level, 37 genera were significantly correlated with at least one parameter. Nine genera, namely *norank_f_Eubacterium_coprostanoligenes_group*, *norank_f_norank_o_Clostridia_UCG-014*, *Ruminococcus*, *norank_f_Gemmatimonadaceae*, *unclassified_f_Lachnospiraceae*, *Prevotellaceae_UCG-001*, *Dechloromonas*, *Lachnospiraceae_NK4A136_group*, and *Marvinbryantia*, were associated with both maternal and fetal parameters (Figure 10). First, we focused on FBG levels and found that *norank_f_Eubacterium_coprostanoligenes_group*, *norank_f_norank_o_Clostridia_UCG-014*, *Ruminococcus*, and *Prevotellaceae_UCG-001* were positively correlated with FBG. Next, we focused on serum biochemical parameters. The nine genera were positively correlated with leptin and negatively correlated with adiponectin. Furthermore, we primarily selected gene-expression indicators related to the development of the fetal nervous system. The nine genera were positively associated with *InsR* and *IGF-1R* mRNA levels and negatively associated with *AKT* mRNA levels. Thus, the correlations between the specific genera of the placenta and maternal and fetal rat parameters may help explain their roles in the development of HIP. Therefore, it can be speculated that J-12 regulates the structure of placental microbiota by mediating maternal-related indicators in rats with HIP, thereby affecting fetal development of the central nervous system.

## 4. Discussion

In the present study, we established an HFD/STZ-induced animal model and comprehensively examined the effect of J-12 intervention on rats and pregnancy outcomes. Subsequently, the correlation between rats and fetuses and the key role of the probiotic strain in this process were further explored through placental microbiota assessment.

### 4.1. J-12 Improved HIP Symptoms, except for FBG, in Mother Rats

The glucose requirements of the host during pregnancy tend to increase, especially during the middle and later stages, which may induce a series of physiological changes, such as insulin resistance, resulting in hyperglycemia and related adverse symptoms. Accumulated evidence has revealed that probiotics have beneficial effects on the prevention and treatment of HIP [34,35,36,37]. For example, *L. rhamnosus* GG and *B. animalis* subsp. *lactis* BB12 can protect against elevated blood glucose levels during pregnancy and reduce the incidence of GDM [34,35]. In pregnant mice receiving an HFD, probiotic supplementation can impact FBG levels and dyslipidemia [37]. However, few studies have examined interventions with a single probiotic strain in HFD/STZ-induced rats with HIP that present mixed type I and II diabetic symptoms. In the present study, intervention with J-12 or insulin alone did not significantly improve FBG levels in rats with HIP because of shorter gestation in rodents (only 21 days). However, the combination of J-12 and insulin significantly reduced FBG levels in late pregnancy, and the intake of J-12 alone improved dyslipidemia, insulin, and leptin resistance (1.00 mIU/L, 9.16 ng/mL) and activated adiponectin (14.97 mg/L) in rats. Insulin resistance caused insufficient leptin secretion [22,38], and the leptin content in the I group and J + I group was significantly lower than that in the CK group, and this significantly affected the fat anabolism in the liver, resulting in a significant increase in triglyceride content. Even the J12 intervention failed to change the outcome. Collectively, these results suggested that J-12 has the potential to prevent HIP.

In addition, J-12 reduced pancreatic and hepatic tissue damage in rats with HIP. The destructive effect of STZ on insulin-producing β-cells has been established based on pancreatic histopathology [39,40]. Consistently, we administered a single STZ injection to induce damage in islet cells, thereby resulting in insufficient insulin secretion. Furthermore, researchers have discovered the regenerative potential and protective effect of Launaea acanthodes extract on islet function by histological observation of pancreatic tissues derived from STZ-induced diabetic rats [41]. DIBc, a nanochelating-based nano-metal–organic framework, reportedly increases the number and area of islets in rats with HFD/STZ-induced diabetes [42]. These findings are consistent with the ameliorative effect of J-12 in the present study. In addition, insulin exhibits antihyperglycemic, anti-inflammatory, and antioxidant effects. Histological morphological observation revealed that insulin treatment relieves islet β cells and promotes cell recovery following hyperglycemia-induced destruction [43]. The present study also demonstrated that intervention with J-12 or insulin improved pancreatic histopathological damage in rats with HIP. 

Several studies have shown that, in hyperglycemic rats, cinnamaldehyde and isoquercitrin can alleviate hepatic tissue lesions, such as unclear liver cell structures and cystic lesions [44,45]. Additionally, Xu et al. investigated the anti-inflammatory activity of puerarin in rats with gestational diabetes induced by HFD/STZ and found that the GDM group showed mild-to-moderate hepatocyte steatosis and inflammatory cell infiltration, as well as degeneration and necrosis of individual hepatocytes [46]. Likewise, our study reported the effects of probiotics on the hepatic histomorphology of rats with HIP. In summary, J-12 promoted the slow recovery of pancreatic tissue and prevented fatty and hepatic functional damage in rats with HIP.

### 4.2. J-12 Improved Physiologic Parameters and Tissue Damage in Fetuses

Evidence regarding fetal skeletal abnormalities in rats with HIP remains in the exploratory phase. For instance, fetal skeletal abnormalities, mainly sternal and caudal abnormalities, were found to occur in diabetic rats exposed to smoke before and during pregnancy [47]. However, there are no reports on the effects of bioactive substances on the skeletal abnormalities of fetuses from HIP mothers. Herein, we reported that J-12 could alleviate the absence of the sternal xiphoid process. A previous study has indicated that insulin resistance could affect the skeletal growth of hyperglycemic rats fed an HFD [48]. The administration of exogenous insulin to rats may affect fetal skeletal development. Therefore, the intervention with insulin and J-12 combined with insulin resulted in the development of sternal cleft, unossified coccygeal vertebra, and incomplete base vertebra, demonstrating that insulin weakened the positive effect of J-12 on the skeletal structure.

Although previous studies have examined the effects of bioactive substances on fetal cardiac, renal, and hepatic morphologies in rats with HIP, few have determined the potential of probiotics. For example, calcitriol and/or pomegranate extract can improve myocardial damage in fetuses [49]. Vitamin E and folic acid supplementation during pregnancy could reduce myocardial cell apoptosis; folic acid had a better inhibitory effect on apoptosis than vitamin E, with superior effects exhibited by the synergistic effect of both [50]. The present study examined fetal cardiac tissue derived from HIP mothers by H&E staining and TEM. We found that insulin, J-12, or a combination of J-12 and insulin could alleviate tissue damage.

HIP can also affect the development of the fetal urinary system, producing congenital renal malformations that typically occur during early organogenesis, such as renal agenesis, hydronephrosis, and ureteral abnormalities [51]. Researchers have shown that insulin improves glomerular morphological abnormalities in the fetuses of diabetic rats [27]. In the current study, H&E staining was used to assess the renal tissue of fetuses from HIP mothers and found that insulin, J-12, or a combination of J-12 and insulin could alleviate tissue damage. Furthermore, our study found that glomerular lesions in the I, A, and J + I groups were attenuated by TEM.

El-Beeh et al. reported that the fetal liver from GDM Wistar rats showed reduced hepatic cords and vascularization, while cold-pressed rosemary oil could alleviate this phenomenon [52]. Similarly, we confirmed that insulin, J-12, or a combination of J-12 and insulin alleviated hepatic morphological damage in the fetus.

### 4.3. J-12 Regulated Gene Expression in the Fetal Central Nervous System 

The hippocampus is a brain structure that is particularly susceptible to changes in glucose concentration, with important behavioral and physiological functions, such as spatial learning and memory, participating in neurodevelopmental and neurocognitive abnormalities in fetuses [53,54,55]. Insulin and IGF-1 are important regulators of central nervous system development and cognitive function [56]. AKT is known to participate in several key signaling pathways in cells and plays an important role in the phosphoinositide-3-kinase (PI3K) signaling pathway, including promotion of glucose metabolism, prevention of gluconeogenesis and hepatic lipidosis formation, and regulation of neuronal proliferation and differentiation [57]. Hami et al. investigated the fetal expression of *InsR* and *IGF-1R* on days 0, 7, and 14 (P0, P7, and P14) in gestational diabetic rats and found that the expression was significantly increased at P0 when compared with the control and insulin groups [56]; this is consistent with our results. Similarly, Tehranipour et al. have shown that STZ-induced maternal diabetes can significantly reduce the density of hippocampal neurons in the 0th generation after birth [58]. Another study has reported that increased hippocampal *IGF-1R* mRNA expression was associated with aging and cognitive decline, suggesting that upregulation of *IGF-1R* mRNA levels may be part of the degenerative atrophy response to the hippocampal formation during aging [59]. Consequently, our results clarified that both J-12 and insulin improved the negative effects on the development of hippocampal nervous system, as well as the cognitive development of fetuses with maternal hyperglycemia; the synergistic effect of J-12 and insulin could further enhance this remission effect. 

Insulin activates the glucose metabolism activity of tissue cells as a signal of glucose richness. First, insulin binds to cell surface receptors and activates the PI3K/AKT pathway through insulin receptor substrate 1 (IRS1). Subsequently, AKT directly promotes glucose absorption and activates downstream pathways (such as mTORC1) to further guide the synthesis of enzymes related to glucose biosynthesis for nutrient storage [60,61,62]. Hami et al. illustrated that the abundance of fetal *AKT* mRNA expression at P0 of GDM rats was significantly lower than that in the normal and insulin-induced groups, and these results are similar to those of the present study. They also found that the abundance of phosphorylated protein expression at different AKT modification sites was consistent with the changing trend of mRNA, indicating that the activation of the AKT pathway was inhibited in GDM rats [57]. However, no report has examined the effect of bioactive substances on the expression of hippocampal-related genes at P0 in the fetus of HIP animal models. Our study indicated that, in the early stages of hippocampal development, both J-12 and insulin could activate the AKT pathway, which may be regulated by the expression of key genes of the PI3K/AKT signaling pathway, thereby participating in the regulation of glucose metabolism and affecting the development of the central nervous system in the fetuses from HIP rats.

### 4.4. J-12 Improved Placental Tissue Damage and Placental Community Diversity in Rats with HIP

Placental function is affected by several factors, including placental morphology [63]. Human and rodent placentas are villous placentas, including syncytiotrophoblast and cytotrophoblast cells, which form the maternal–fetal barrier of villous tissue, thus effectively regulating the transfer and circulation of nutrients from mother to fetus [33]. By establishing a rat GDM model, a recent study has demonstrated that cinnamaldehyde alleviates placental structural abnormalities [25]. However, the effects of probiotics in this area remain unknown. The present study demonstrated that J-12 alleviates placental structural abnormalities.

Furthermore, the present study analyzed the placental microbiota of HIP rats and found that certain placental microorganisms were associated with positive effects on fetuses and HIP mother rats supplemented with J-12. Thus, we analyzed changes in the structure and diversity of the placental microbiota. Consistent with previous studies [18,64], non-pathogenic symbiotic microorganisms were mainly present in the placenta, with Proteobacteria found to be the most abundant. However, we found that the relative levels of Pachomycetes decreased in the placenta of rats with HIP after J-12 intervention, and the relative levels of Actinomycetes were increased after intervention with J-12 combined with insulin. To date, no similar studies have been reported. A few previous reports on placental microbiota are available [19,46,65,66,67,68,69], among which clinical studies on the placental microbiota of HIP remain at phylum and genus levels [65,67,69], and studies related to basic animal experiments with HIP are lacking. 

Herein, we analyzed the placental microbiota of rats with HIP, mainly at the genus and species levels. In a case study of respiratory tract infection in premature infants, *Delftia_tsuruhatensis* was found in patients with immune system damage [70]. This may be attributed to the unbalanced state of the immune system of rats with HIP [71], which is similar to immune system damage, thereby resulting in similar dominant species. *Weissella_cibaria* is a lactic acid bacterium, and Lee et al. have shown that its biotransformed whey may exert an anti-lipogenic effect by inhibiting the signaling events of intracellular fat-related transcription factors and target genes [72]. We found that the M group had a relatively high proportion of *Weissella_cibaria* when compared with other groups, probably because the placenta showed maladaptive changes through this self-regulation, and J-12 could improve this unfavorable environment to some extent. A review by Nakayama revealed that *Corynebacterium_glutamicum* could secrete L-glutamate through mechanically sensitive channels in the intestinal microbiota, which regulates host brain function via the microbiota–gut–brain axis [73]. Herein, it can be suggested that J-12 intervention of the placental microbiota may affect the brain function of fetuses via the microbiota–placental–brain axis. In addition, this study predicted the metabolic profile of the placental microbiome after J-12 consumption. Interestingly, J-12 could regulate the metabolism and genetic- and environmental-information-processing-related pathways in rats with HIP, indicating that J-12 may regulate the placental microbiota in a variety of ways to improve HIP. However, because of the complexity of the placental microbiota, the precise regulatory mechanism of J-12 needs to be further elucidated.

### 4.5. Correlation between Overall Placental Microbiota Structure and Maternal and Fetal Rats Parameters

In a previous study, Ruminococcus was found to be involved in the phosphorylation of cellobiose and glucose [74]. In a study exploring the relationship between GDM and placental- and umbilical-cord-blood microbiota, Tang et al. reported elevated glucose levels in GDM, accompanied by a greater abundance of Ruminococcus in the placenta [69]. Ibrahim et al. revealed that Prevotellaceae_UCG-001 can affect immune system diseases and metabolic functions [75]. In the present study, we found that these two genera were enriched in the placenta of rats with HIP. Thus, we speculated that HIP could impact the host’s immune system and metabolic functions by affecting the placental microbiota. The roles of the other seven genera, including *norank_f_Eubacterium_coprostanoligenes_group*, *norank_f_norank_o_Clostridia_UCG-014*, *norank_f_Gemmatimonadaceae*, *unclassified_f_Lachnospiraceae*, *Dechloromonas*, *Lachnospiraceae_NK4A136_group*, and *Marvinbryantia*, were not elaborated in the relevant areas of this study. In addition, these genera were positively correlated with FBG and leptin levels in rats and hippocampal levels of *InsR* and *IGF-1R* mRNA in fetuses. In contrast, they were negatively correlated with adiponectin levels in rats and hippocampal levels of *AKT* mRNA in fetuses. However, no previous studies have reported the effects of bioactive substances on the placental microbiota in HIP. Therefore, it was speculated that J-12 regulates the structure of placental microbiota by mediating maternal-related indicators in rats with HIP, thereby affecting the development of the fetal central nervous system. However, this is only a preliminary discussion, and the specific causal relationship requires further verification.

Current clinical trials of probiotics for the treatment of disorders of glucolipid metabolism and gestational diabetes mellitus (GDM) have also been reported. Several studies indicated that probiotics treatment may reduce the HbA1c, FBG, and insulin-resistance levels in T2DM patients and revealed that lipids, blood pressure, and inflammation indicators are significantly improved by probiotic supplementation [76,77,78]. Wickens et al. have found that *Lactobacillus rhamnosus* HN001 supplementation from 14 to 16 weeks’ gestation may reduce gestational diabetes mellitus (GDM) prevalence, particularly among older women and those with previous GDM [13]. *Lactobacillus rhamnosus* GG, *Bifidobacterium bifidum*, *Lactobacillus acidophilus*, *Lactobacillus casei*, and *Lactobacillus fermentum* supplementation in patients with GDM had beneficial effects on gene expression related to insulin and inflammation, glycemic control, few lipid profiles, inflammatory markers, and oxidative stress [35,79]. These results have shown that probiotic-supplemented perinatal dietary counselling could be a safe and cost-effective tool in addressing the metabolic epidemic. However, few animal studies and clinical trials have evaluated the effects of probiotics on the offspring of HIP, and a further evaluation of the effects of probiotics on the offspring remains to be completed, including clinical and animal studies. Therefore, based on previous studies, our results may support another novel idea that probiotics can alleviate HIP and have a beneficial effect on offspring, an idea that could be explored further in clinical trials.

## 5. Conclusions

Herein, J-12 improved dyslipidemia, insulin and leptin resistance, activated adiponectin, and reduced pancreatic and hepatic tissue damage in rats with HIP. Furthermore, J-12 alleviated fetal skeletal abnormalities; reduced cardiac, renal, and hepatic damage; and regulated gene expression in the fetal central nervous system. Finally, J-12 may affect the development of the fetal central nervous system via the placental microbiota by regulating maternal-related indicators. Our findings provide novel insights into the treatment of HIP, but the precise causality needs to be verified by undertaking in-depth basic assessments and clinical trials.

## Figures and Tables

**Figure 1 nutrients-15-00170-f001:**
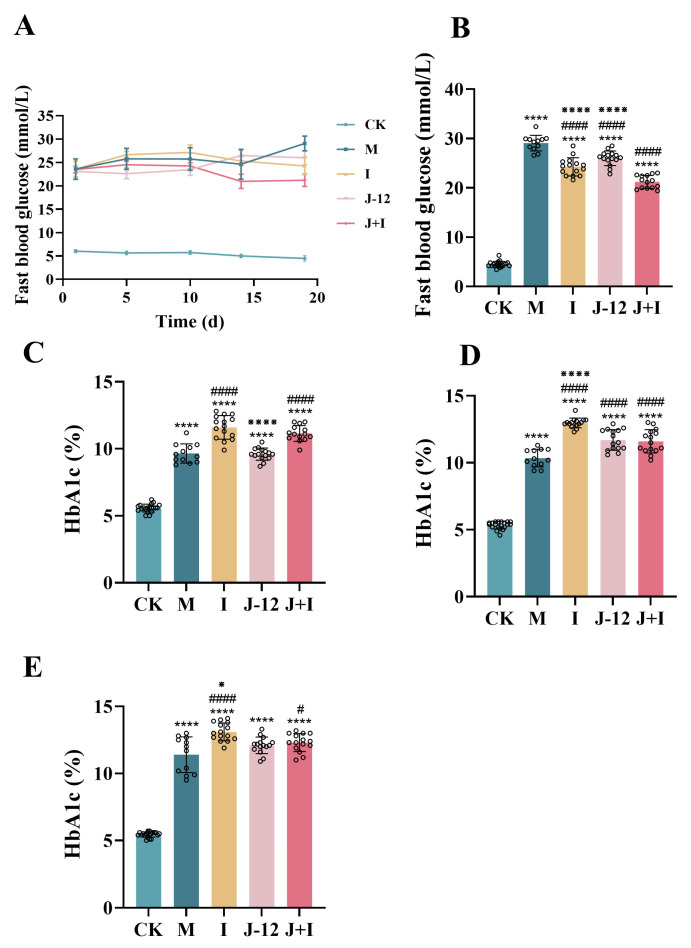
Fasting blood glucose and HbA1c levels during pregnancy in rats. (**A**) Changes in fasting blood during pregnancy. (**B**) Fasting blood glucose in the late of pregnancy (GD 19). (**C**) HbA1c levels in the start of pregnancy (GD 1). (**D**) HbA1c levels in the middle of pregnancy (GD 10). (**E**) HbA1c levels in the late of pregnancy (GD 19). CK: control group, rats fed a standard diet; *n* = 20. M: model group, rats fed an HFD and administered a single STZ injection prior to pregnancy and then fed a standard diet during pregnancy; *n* = 12. I: insulin group, rats fed an HFD and administered a single STZ injection prior to pregnancy and then fed a standard diet and injected the insulin and during pregnancy; *n* = 15. J-12: J-12 group, rats fed an HFD and administered a single STZ injection prior to pregnancy and then fed a standard diet during pregnancy. Rats administered the J-12 suspension (1 × 10^9^ CFU daily) by gavage during the entire experimental period; *n* = 15. J + I: J-12 plus insulin group, rats fed an HFD and administered a single STZ injection prior to pregnancy and then fed a standard diet and injected with insulin during pregnancy. Rats administered the J-12 suspension (1 × 10^9^ CFU daily) by gavage during the entire experimental period; *n* = 14. Statistical analyses were performed by using a one-way ANOVA with Tukey’s multiple comparison test. Data are presented as the means ± SD; **** *p* < 0.0001, compared with the CK group; ^#^
*p* < 0.05 and ^####^
*p* < 0.0001, compared with the M group; ^※^
*p* < 0.05 and ^※※※※^
*p* < 0.0001, compared with the J + I group.

**Figure 2 nutrients-15-00170-f002:**
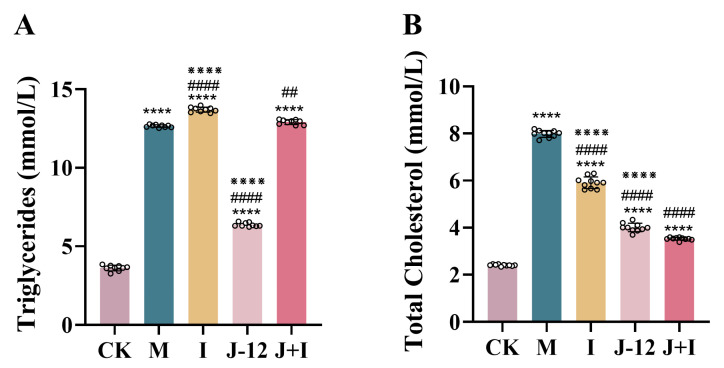
J-12 improved dyslipidemia in rats with hyperglycemia in pregnancy. (**A**) Triglycerides. (**B**) Total cholesterol. CK: control group, rats fed a standard diet. M: model group, rats fed an HFD and administered a single STZ injection prior to pregnancy and then fed a standard diet during pregnancy. I: insulin group, rats fed an HFD and administered a single STZ injection prior to pregnancy and then fed a standard diet and injected the insulin and during pregnancy. J-12: J-12 group, rats fed an HFD and administered a single STZ injection prior to pregnancy and then fed a standard diet during pregnancy. Rats administered the J-12 suspension (1 × 10^9^ CFU daily) by gavage during the entire experimental period. J + I: J-12 plus insulin group, rats fed an HFD and administered a single STZ injection prior to pregnancy and then fed a standard diet and injected with insulin during pregnancy. Rats administered the J-12 suspension (1 × 10^9^ CFU daily) by gavage during the entire experimental period; *n* = 9 per group. Statistical analyses were performed by using a one-way ANOVA with Tukey’s multiple comparison test. Data are presented as the means ± SD; **** *p* < 0.0001, compared with the CK group; ^##^
*p* < 0.01 and ^####^
*p* < 0.0001, compared with the M group; ^※※※※^
*p* < 0.0001, compared with the J + I group.

**Figure 3 nutrients-15-00170-f003:**
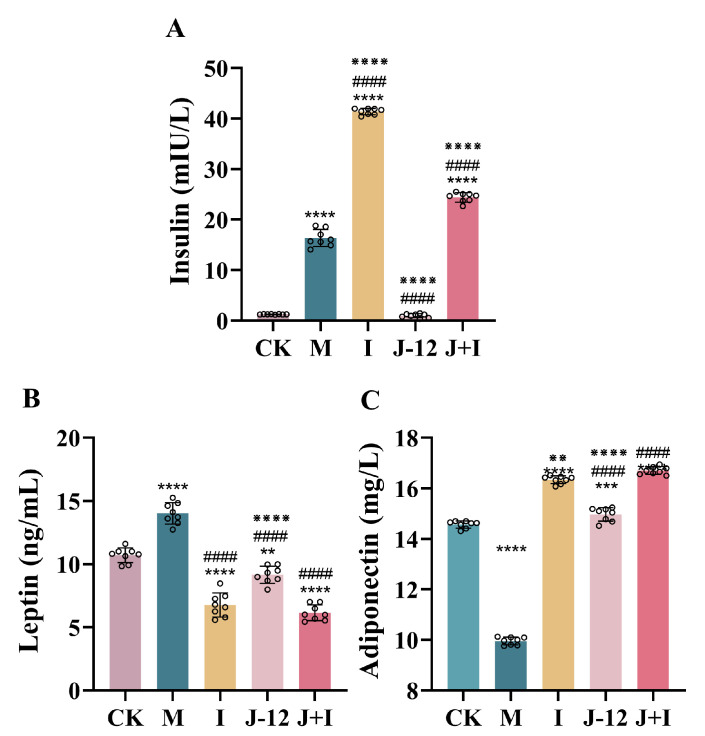
J-12 improved insulin and leptin resistance and activated adiponectin in rats with hyperglycemia in pregnancy. (**A**) Serum insulin levels. (**B**) Serum leptin levels. (**C**) Serum adiponectin levels. CK: control group, rats fed a standard diet. M: model group, rats fed an HFD and administered a single STZ injection prior to pregnancy and then fed a standard diet during pregnancy. I: insulin group, rats fed an HFD and administered a single STZ injection prior to pregnancy and then fed a standard diet and injected with insulin during pregnancy. J-12: J-12 group, rats fed an HFD and administered a single STZ injection prior to pregnancy and then fed a standard diet during pregnancy. Rats administered the J-12 suspension (1 × 10^9^ CFU daily) by gavage during the entire experimental period. J + I: J-12 plus insulin group, rats fed an HFD and administered a single STZ injection prior to pregnancy and then fed a standard diet and injected with insulin during pregnancy. Rats administered the J-12 suspension (1 × 10^9^ CFU daily) by gavage during the entire experimental period; *n* = 8 per group. Statistical analyses were performed by using a one-way ANOVA with Tukey’s multiple comparison test. Data are presented as the means ± SD; ** *p* < 0.01, *** *p* < 0.001, and **** *p* < 0.0001, compared with the CK group; ^####^
*p* < 0.0001, compared with the M group; ^※※^
*p* < 0.01 and ^※※※※^
*p* < 0.0001, compared with the J + I group.

**Figure 4 nutrients-15-00170-f004:**
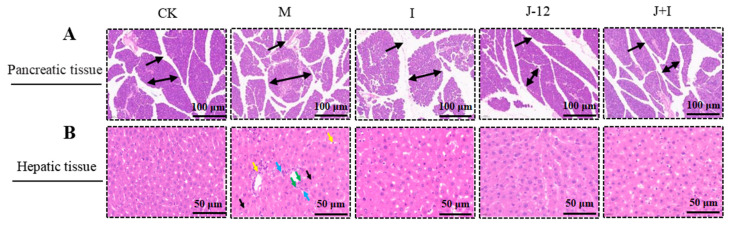
J-12 reduced pancreatic and hepatic tissue damage in rats with hyperglycemia in pregnancy. (**A**) H&E staining in pancreatic sections. Islet cells are indicated by the double arrow, and interlobular catheter are indicated by the single arrow. Magnification = ×100. Scale bar: 100 μm. (**B**) H&E staining in hepatic sections. Liver cells are indicated by the black arrow, inflammatory cells are indicated by the yellow arrow, fibrous connective tissue cells are indicated by the blue arrow, and bile duct cells are indicated by the green arrow. CK: control group, rats fed a standard diet. M: model group, rats fed an HFD and administered a single STZ injection prior to pregnancy and then fed a standard diet during pregnancy. I: insulin group, rats fed an HFD and administered a single STZ injection prior to pregnancy and then fed a standard diet and injected with insulin during pregnancy. J-12: J-12 group, rats fed an HFD and administered a single STZ injection prior to pregnancy, then, fed a standard diet during pregnancy. Rats administered the J-12 suspension (1 × 10^9^ CFU daily) by gavage during the entire experimental period. J + I: J-12 plus insulin group, rats fed an HFD and administered a single STZ injection prior to pregnancy, then, fed a standard diet and injected the insulin and during pregnancy. Rats administered the J-12 suspension (1 × 10^9^ CFU daily) by gavage during the entire experimental period. Magnification = ×400. Scale bar: 50 μm.

**Figure 5 nutrients-15-00170-f005:**
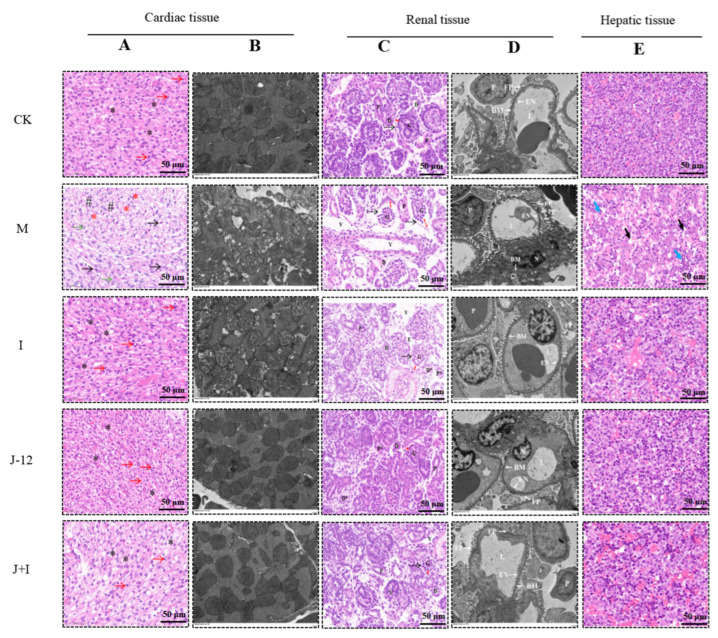
J-12 reduced cardiac, renal, and hepatic tissue damage of fetuses. (**A**) H&E staining in cardiac sections. Mitosis is indicated by the red arrows, and transverse band of cardiomyocytes is indicated by “*”. Cellular nuclear sequestration is indicated by the black arrow, cytoplasmic hydrographic changes is indicated by “#”, chromatin hydrolysis of the nucleus is indicated by the green arrows, and cytoplasmic vacuolization is indicated by the red “*”. Magnification = ×400. Scale bar: 50 μm. (**B**) Transmission electron microscopy (TEM) in cardiac sections. Magnification = ×5.0 k. Scale bar: 2 μm. (**C**) H&E staining in renal sections. Renal capsule is indicated by the dashed arrow, erythrocyte space is indicated by the double arrow. Glomerular (G), proximal convoluted tubules (P/P*), distal convoluted tubules (D/D*), tubular vacuole (V). Magnification = ×400. Scale bar: 50 μm. (**D**) Transmission electron microscopy (TEM) in renal sections. Glomerular capillary lumen (L), endothelial cell with euchromatic nucleus (N), fenestrated layer (EN), basement membrane (BM), podocyte (P), foot processes (FP). Magnification = ×3.0 k. Scale bar: 5 μm. (**E**) H&E staining in hepatic sections. Hepatic cord rupture is indicated by the black arrow, and relative increase of hematopoietic cells is indicated by the blue arrow. CK: control group, rats fed a standard diet. M: model group, rats fed an HFD and administered a single STZ injection prior to pregnancy and then fed a standard diet during pregnancy. I: insulin group, rats fed an HFD and administered a single STZ injection prior to pregnancy and then fed a standard diet and injected the insulin and during pregnancy. J-12: J-12 group, rats fed an HFD and administered a single STZ injection prior to pregnancy and then fed a standard diet during pregnancy. Rats administered the J-12 suspension (1 × 10^9^ CFU daily) by gavage during the entire experimental period. J + I: J-12 plus insulin group, rats fed an HFD and administered a single STZ injection prior to pregnancy and then fed a standard diet and injected with insulin during pregnancy. Rats administered the J-12 suspension (1 × 10^9^ CFU daily) by gavage during the entire experimental period. Magnification = ×400. Scale bar: 50 μm.

**Figure 6 nutrients-15-00170-f006:**
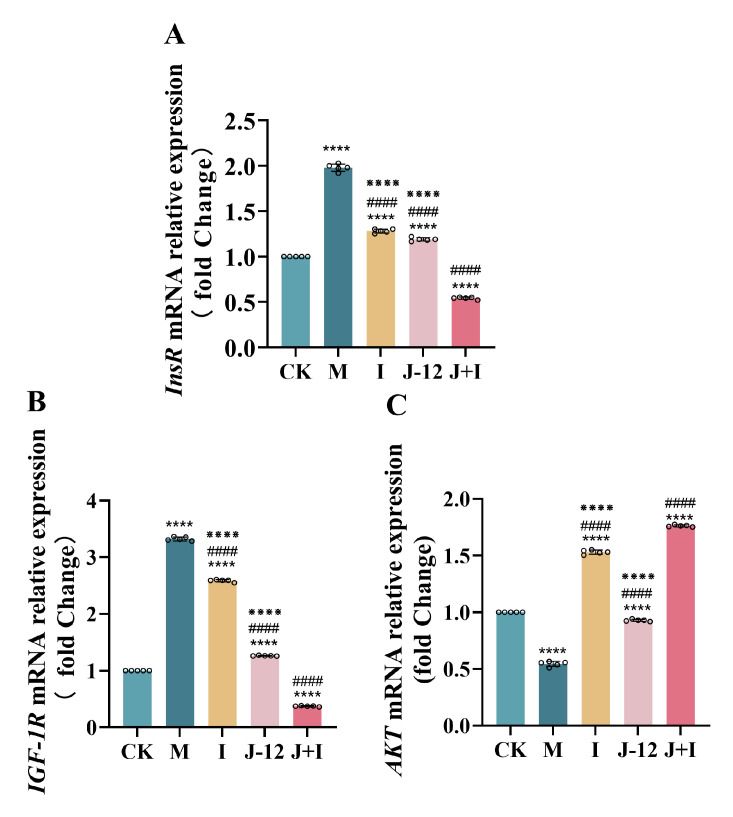
J-12 regulated gene expression in the central nervous system of fetuses. (**A**) Insulin receptor (*InsR*) mRNA relative expression in hippocampus. (**B**) Insulin-like growth factor-1 receptor (*IGF-1R*) mRNA relative expression in hippocampus. (**C**) Protein kinase B (*AKT*) mRNA relative expression in hippocampus. CK: control group, rats fed a standard diet. M: model group, rats fed an HFD and administered a single STZ injection prior to pregnancy and then fed a standard diet during pregnancy. I: insulin group, rats fed an HFD and administered a single STZ injection prior to pregnancy and then fed a standard diet and injected with insulin during pregnancy. J-12: J-12 group, rats fed an HFD and administered a single STZ injection prior to pregnancy and then fed a standard diet during pregnancy. Rats administered the J-12 suspension (1 × 10^9^ CFU daily) by gavage during the entire experimental period. J + I: J-12 plus insulin group, rats fed an HFD and administered a single STZ injection prior to pregnancy and then fed a standard diet and injected with insulin during pregnancy. Rats administered the J-12 suspension (1 × 10^9^ CFU daily) by gavage during the entire experimental period; *n* = 5 per group. Statistical analyses were performed by using a one-way ANOVA with Tukey’s multiple comparison test. Data are presented as the means ± SD; **** *p* < 0.0001, compared with the CK group; ^####^
*p* < 0.0001, compared with the M group; ^※※※※^
*p* < 0.0001, compared with the J + I group.

**Figure 7 nutrients-15-00170-f007:**
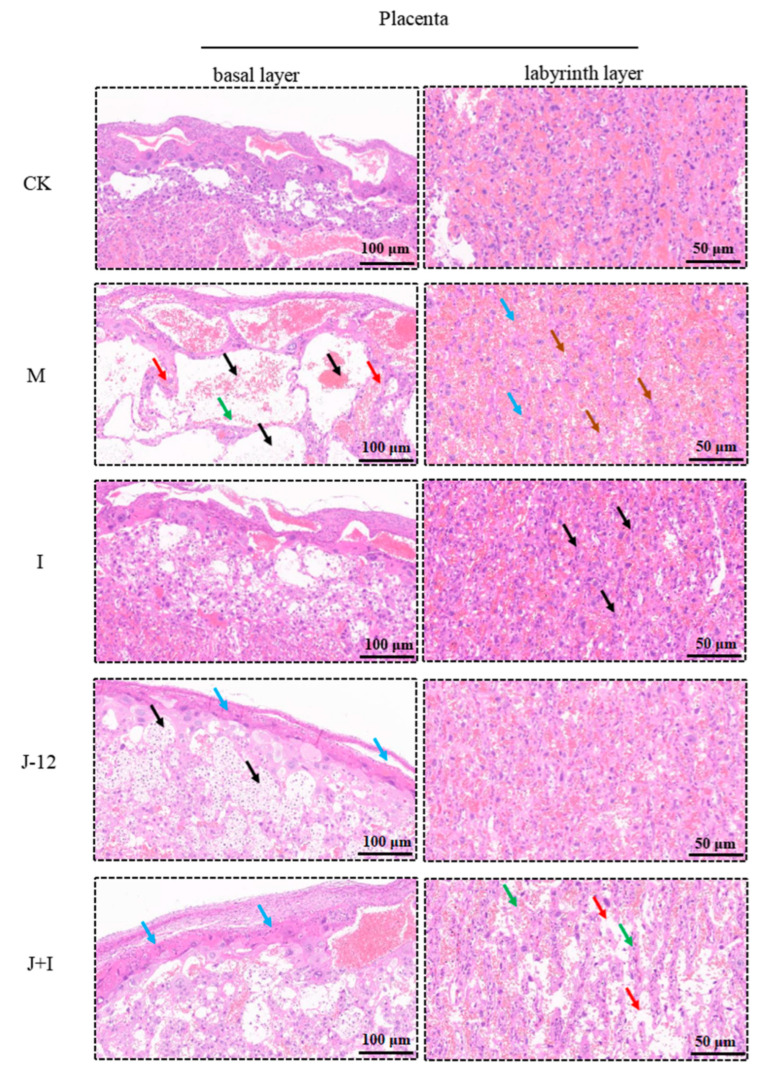
J-12 reduced placental tissue damage in rats with hyperglycemia in pregnancy. H&E staining in placental sections. M group: Cystic degeneration of glycogen cells is indicated by the black arrow, and edema and degeneration of sponge trophoblast cells are indicated by the green and red arrow, respectively, in the basal layer. Villi atrophy and fragmentation are indicated by the blue arrow, and fetal vascular reduction is indicated by the brown arrow in labyrinth layer. I group: fetal vascular reduction as indicated by the black arrow in the labyrinth layer. J-12 group: Glycogen cells are indicated by the black arrow, and cytolysis of giant cells is indicated by the blue arrow in the basal layer. J + I group: Cytolysis of giant cells is indicated by the blue arrow, and edema of sponge trophoblast cells is indicated by the brown arrow in basal layer. Villi fragmentation is indicated by the blue arrow, and aqueous degeneration of trophoblasts is indicated by the green arrow in the labyrinth layer. CK: control group, rats fed a standard diet. M: model group, rats fed an HFD and administered a single STZ injection prior to pregnancy and then fed a standard diet during pregnancy. I: insulin group, rats fed an HFD and administered a single STZ injection prior to pregnancy and then fed a standard diet and injected with insulin during pregnancy. J-12: J-12 group, rats fed an HFD and administered a single STZ injection prior to pregnancy and then fed a standard diet during pregnancy. Rats administered the J-12 suspension (1 × 10^9^ CFU daily) by gavage during the entire experimental period. J + I: J-12 plus insulin group, rats fed an HFD and administered a single STZ injection prior to pregnancy and then fed a standard diet and injected with insulin during pregnancy. Rats administered the J-12 suspension (1 × 10^9^ CFU daily) by gavage during the entire experimental period. Basal layer: magnification = ×100, and scale bar = 100 μm. Labyrinth layer: magnification = ×400, and scale bar = 50 μm.

**Figure 8 nutrients-15-00170-f008:**
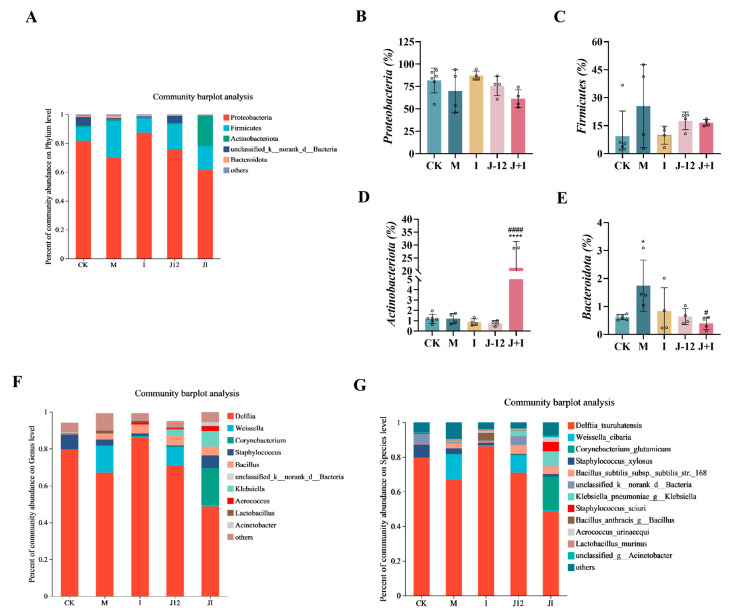
J-12 altered placental community diversity in rats with hyperglycemia in pregnancy. (**A**) Community structure of the placental microbiota at the phylum level. (**B**–**E**) Relative abundances of Proteobacteria, Firmicutes, Actinobacteriota, and Bacteroidota. Statistical analyses were performed by using a one-way ANOVA with Tukey’s multiple comparison test. Data are presented as the means ± SD; * *p* < 0.05 and **** *p* < 0.0001, compared with the CK group; ^#^
*p* < 0.05 and ^####^
*p* < 0.0001, compared with the M group. (**F**) Community structure of the placental microbiota at the genus level. (**G**) Community structure of the placental microbiota at the species level. (**H**) Multilevel discrimination map of species differences in LEfSe. CK: control group, rats fed a standard diet. M: model group, rats fed an HFD and administered a single STZ injection prior to pregnancy and then fed a standard diet during pregnancy. I: insulin group, rats fed an HFD and administered a single STZ injection prior to pregnancy and then fed a standard diet and injected with insulin during pregnancy. J-12: J-12 group, rats fed an HFD and administered a single STZ injection prior to pregnancy and then fed a standard diet during pregnancy. Rats administered the J-12 suspension (1 × 10^9^ CFU daily) by gavage during the entire experimental period. J + I: J-12 plus insulin group, rats fed an HFD and administered a single STZ injection prior to pregnancy and then fed a standard diet and injected with insulin during pregnancy. Rats administered the J-12 suspension (1 × 10^9^ CFU daily) by gavage during the entire experimental period. The threshold of logarithmic LDA score is 3.5.

**Figure 9 nutrients-15-00170-f009:**
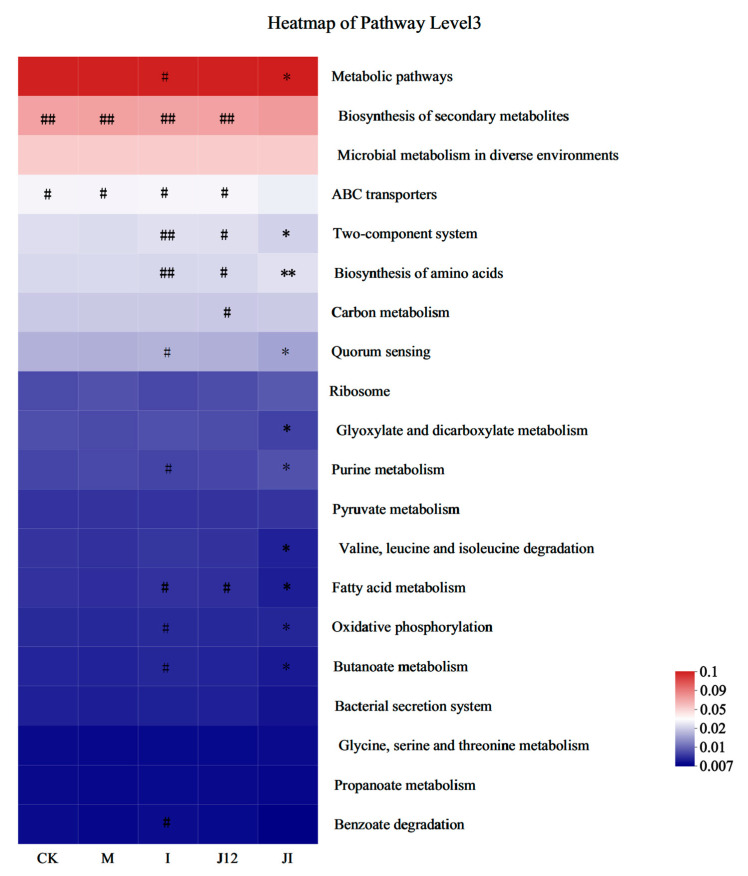
Predicted metabolic profile of the placental microbiome after J-12 supplementation. The 16S rRNA data were further analyzed as indicated by the PICRUSt. Statistical analyses were performed by using a one-way ANOVA with Tukey’s multiple comparison test. Data are presented as the means ± SD; * *p* < 0.05, ** *p* < 0.01, compared with the CK group; ^#^
*p* < 0.01 and ^##^
*p* < 0.01, compared with the J + I group.

**Figure 10 nutrients-15-00170-f010:**
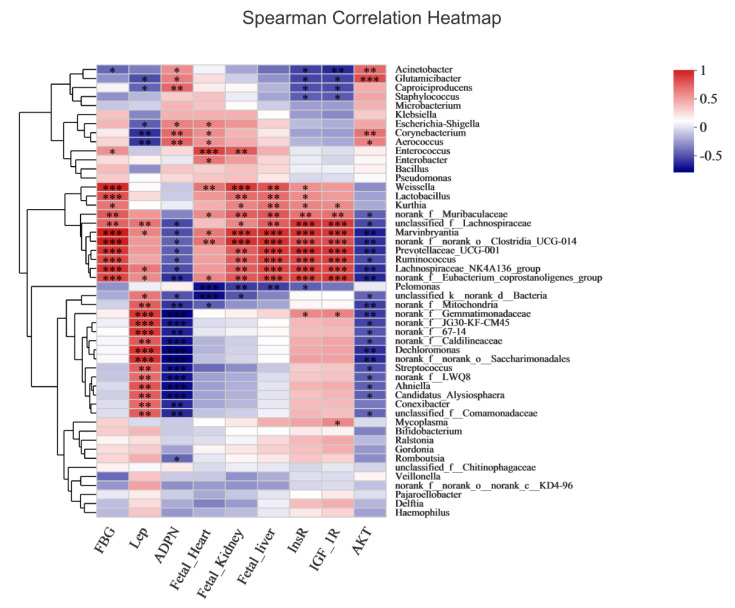
Correlation of placental microbiota structure with maternal and fetal rats’ parameters by Spearman correlation heatmap. Fasting blood glucose (FBG), HbA1c levels (HbA1c), triglycerides (TG), total cholesterol (TCHO), insulin (Ins), leptin (Lep), adiponectin (ADPN), insulin receptor (InsR), insulin-like growth factor-1 receptor (IGF-1R), and protein kinase B (AKT); * *p* < 0.05, ** *p* < 0.01, and *** *p* < 0.001.

**Table 1 nutrients-15-00170-t001:** Oligonucleotide primers used to amplify RNA transcripts.

Primers Name	Oligonucleotide (5′-3′)	Oligonucleotide (3′-5′)	Tm (°C)
*InsR*	GGCCCGATGCTGAGAACA	CGTCATTCCACGTCATTCCA	60
*IGF-1R*	GCCGTGCTGTGCCTGTCCTAAAAC	GCTACCGTGGTGTTCCTGCTTCG	60
*AKT*	GAAGCTGAGCCCACCTTTCA	CATCTTGATCAGGCGGTGTG	56
*GADPH*	AACTCCCATTCTTCCACCTTTG	CTGTAGCCATATTCATTGTCATACCAG	60
*β-actin*	TATCGGCAATGAGCGGTTCC	ACTGTGTTGGCATAGAGG	56

**Table 2 nutrients-15-00170-t002:** Effect of J-12 on physiologic parameters of fetuses.

Variables	Live Fetuses (%)	Dead Fetuses (%)	Resorptions (%)	Body Weight (g)
CK	100 ^a^	0 ^b^	0 ^b^	3.85 ± 0.09
M	73.8 ^b^	26.2 ^a^	0 ^b^	3.49 ± 0.11
I	97.4 ^a^	0 ^b^	2.6 ^a^	3.39 ± 0.04
J-12	100 ^a^	0 ^b^	0 ^b^	3.64 ± 0.14
J + I	95.2 ^a^	0 ^b^	4.8 ^a^	3.42 ± 0.09

A significance analysis of percentages was performed by using the chi-square test. CK: control group, rats fed a standard diet. M: model group, rats fed an HFD and administered a single STZ injection prior to pregnancy and then fed a standard diet during pregnancy. I: insulin group, rats fed an HFD and administered a single STZ injection prior to pregnancy and then fed a standard diet and injected with insulin during pregnancy. J-12: J-12 group, rats fed an HFD and administered a single STZ injection prior to pregnancy and then fed a standard diet during pregnancy. Rats administered the J-12 suspension (1 × 10^9^ CFU daily) by gavage during the entire experimental period. J + I: J-12 plus insulin group, rats fed an HFD and administered a single STZ injection prior to pregnancy and then fed a standard diet and injected with insulin during pregnancy. Rats administered the J-12 suspension (1 × 10^9^ CFU daily) by gavage during the entire experimental period. Data are presented as the means ± SD. Shoulder marked with the different letter indicates that the difference was significant *p* < 0.05.

**Table 3 nutrients-15-00170-t003:** Effect of J-12 on physiologic parameters of fetuses.

Variables	CK	M	I	J-12	J + I
Absence of the xiphoid process of the sternum (%)	0 ^c^	100 ^a^	100 ^a^	25 ^b^	80 ^a^
Sternum fractur (%)	0 ^b^	0 ^b^	50 ^a^	0 ^b^	0 ^b^
Unossified coccyx (%)	0 ^b^	0 ^b^	0 ^b^	0 ^b^	80 ^a^
Incomplete ossification of the bottom vertebra (%)	0 ^b^	0 ^b^	0 ^b^	0 ^b^	20 ^a^

A significance analysis of percentages was performed by using the chi-square test. CK: control group, rats fed a standard diet. M: model group, rats fed an HFD and administered a single STZ injection prior to pregnancy and then fed a standard diet during pregnancy. I: insulin group, rats fed an HFD and administered a single STZ injection prior to pregnancy and then fed a standard diet and injected the insulin and during pregnancy. J-12: J-12 group, rats fed an HFD and administered a single STZ injection prior to pregnancy and then fed a standard diet during pregnancy. Rats administered the J-12 suspension (1 × 10^9^ CFU daily) by gavage during the entire experimental period. J + I: J-12 plus insulin group, rats fed an HFD and administered a single STZ injection prior to pregnancy and then fed a standard diet and injected the insulin and during pregnancy. Rats administered the J-12 suspension (1 × 10^9^ CFU daily) by gavage during the entire experimental period; *n* = 6 per group. Shoulder marked with the different letter indicates that the difference was significant at *p* < 0.05.

**Table 4 nutrients-15-00170-t004:** The alpha diversity of placental microbiota in rats with hyperglycemia in pregnancy at the genus level.

Variables	CK	M	I	J-12	J + I	*p*-Value (CK-M)	*p*-Value (M-J-12)	*p*-Value (J-12-J + I)
sobs	202 ± 23.82	227 ± 30.90	177 ± 14.73	136 ± 26.15	96 ± 41.69	0.215	0.004 **	0.153
shannon	0.84 ± 0.21	1.37 ± 0.74	0.74 ± 0.25	1.16 ± 0.33	1.64 ± 0.27	0.167	0.624	0.066
simpson	0.73 ± 0.09	0.52 ± 0.29	0.75 ± 0.09	0.54 ± 0.14	0.34 ± 0.13	0.192	0.968	0.079
ace	223 ± 25.89	272 ± 40.37	217 ± 22.57	159 ± 39.58	106 ± 44.20	0.060	0.007 **	0.120
chao	219 ± 25.27	272 ± 38.62	215 ± 18.44	161 ± 46.82	104 ± 44.37	0.041 *	0.012 *	0.125

CK: control group, rats fed a standard diet. M: model group, rats fed an HFD and administered a single STZ injection prior to pregnancy and then fed a standard diet during pregnancy. I: insulin group, rats fed an HFD and administered a single STZ injection prior to pregnancy and then fed a standard diet and injected with insulin during pregnancy. J-12: J-12 group, rats fed an HFD and administered a single STZ injection prior to pregnancy and then fed a standard diet during pregnancy. Rats administered the J-12 suspension (1 × 10^9^ CFU daily) by gavage during the entire experimental period. J + I: J-12 plus insulin group, rats fed an HFD and administered a single STZ injection prior to pregnancy and then fed a standard diet and injected with insulin during pregnancy. Rats administered the J-12 suspension (1 × 10^9^ CFU daily) by gavage during the entire experimental period. * *p* < 0.05 and ** *p* < 0.01.

## Data Availability

The data presented in this study are available upon request from the authors.

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
