# Peer review of "Ameliorative Effects of Bifidobacterium animalis subsp. lactis J-12 on Hyperglycemia in Pregnancy and Pregnancy Outcomes in a High-Fat-Diet/Streptozotocin-Induced Rat Model"

_nutrients, 2022, doi:10.3390/nu15010170_

Round 1

Reviewer 1 Report

The paper by de Jianjun Yang et al tells about effect of Bifidobacterium animalis on hyperglycemia in pregnancy and pregnancy outcomes in 3 a high-fat diet/streptozotocin-induced rat model.

The idea is publication-worth, the paper is well written, experiments are well done and data are well discussed. I suggest to accept the paper with minor revision of the figures to make them more representative. 

Figs: please add the details regarding treatment groups. e.g. what means CK, M, I etc. The statistically significant difference should be shown by parenthesises. What means abc over the bars?

Metagenomic data: please show statistically significant differences and diversity indices

Author Response

Dear Reviewer:

Reviewer 2 Report

The original manuscript entitled “Improvement Effect of Bifidobacterium animalis subsp. lactis J-12 on Hyperglycemia in Pregnancy and Pregnancy Outcomes in a High-Fat Diet/Streptozotocin-Induced Rat Model” by Jianjun Yang et al. reported the effects of Bifidobacterium-based probiotics on pregnancy outcomes and hyperglycemia in a rodent model. Specifically, the authors chemically induced in a rat model pathophenotypes characteristic of hyperglycemia in pregnancy (HIP), treated pregnant rats with J-12, a type of Bifidobacterium-based probiotics, and determined both the viability of fetuses and concentration levels of maternal circulating markers related to HIP and/or neuropathological outcomes. Placental gut microbiota was also examined for potential changes and functional roles. A protective effect was observed resulting from combined use of J-12 and insulin which failed with each applied individually, as shown in molecular panels of TG/cholesterol levels, insulin/leptin resistance, activated adiponectin as well as morphology of maternal pancreas and hepatic tissues. Maternal microbiota was extensively changed as associated with HIP outcomes. Overall, the manuscript was well written, and results shall be useful for the detailed molecular characterization of Bifidobacterium-induced effects, especially for fetal nervous systems. Yet, there are a few places that need revision or elaboration, as detailed below. Here, I recommend major revision of the manuscript before considering it for acceptance to be published in Nutrients.

Major Comments:

1. Title. The word “improvement effect” is not idiomatic. Maybe change it to “ameliorative effects” or “protective effects.“ Also, italicize “Bifidobacterium animalis subsp. lactis J-12,” if at all possible.

2. Abstract. The rationale is not clear on using Bifidobacterium-based probiotics as a promising therapeutic option treating hyperglycemia in pregnancy and further, why J-12 was chosen as a model for the experiment. I find the first few sentences rather untoward and incoherent, please revise.

3. How did Bifidobacteria change upon feeding Bifidobacterium-based probiotics?

4. The authors observed interesting changes in maternal microbiota under different treatment. What are mechanistic pathways that resident microbiota may contribute to the pregnancy outcomes, other than microbial composition and taxonomic signatures? It would be interesting to perform some predictive pathway analyses on the molecular level such as PICRUSt based on the 16S rRNA data.

5. Overall, how would these rat findings translate to human studies? Please add in the Discussion and list current (or potential) clinical trials that attempted to employ probiotics to mitigate metabolic syndromes.

Minor Comments:

Figures -Show individual points in all bar plots.

Figures -For showing statistical significance, asterisks would usually be more straightforward and better compared with the letters perched atop bar graphs in the original manuscript.

Figures must be intelligible without reference to the text, please at least extend in figure legends to specify full names of each group, especially when acronyms are used in the figures.

Line 273. Figure 2A. It is interesting to see that J-12 reduced TG the most but not so much when insulin was co-applied, which was a totally different case of cholesterol. Add comments in Discussion to explain this.

Line 480. Table 4 contains Chinese characters, please correct.

Line 540. Delete “Authors.”

Author Response

Dear Reviewer,

Round 2

Reviewer 2 Report

I would like to commend and congratulate the authors' dedicated efforts in improving this manuscript. All my concerns have been addressed. Therefore, I recommend the acceptance of the revised manuscript in the present form.